# The Role of Signaling Pathways of Inflammation and Oxidative Stress in Development of Senescence and Aging Phenotypes in Cardiovascular Disease

**DOI:** 10.3390/cells8111383

**Published:** 2019-11-04

**Authors:** John Papaconstantinou

**Affiliations:** 5.138B Medical Research Building, 301 University Boulevard, Department of Biochemistry and Molecular Biology, The University of Texas Medical Branch, Galveston, TX 77555-0643, USA; jpapacon@utmb.edu; Tel.: +1-409-772-2761; Fax: +1-409-772-9216

**Keywords:** cardiomyopathies, senescence, aging, oxidative stress, inflammation, mitochondrial dysfunction, ASK1-signalosome, p38 MAPK

## Abstract

The ASK1-signalosome→p38 MAPK and SAPK/JNK signaling networks promote senescence (in vitro) and aging (in vivo, animal models and human cohorts) in response to oxidative stress and inflammation. These networks contribute to the promotion of age-associated cardiovascular diseases of oxidative stress and inflammation. Furthermore, their inhibition delays the onset of these cardiovascular diseases as well as senescence and aging. In this review we focus on whether the (a) ASK1-signalosome, a major center of distribution of reactive oxygen species (ROS)-mediated stress signals, plays a role in the promotion of cardiovascular diseases of oxidative stress and inflammation; (b) The ASK1-signalosome links ROS signals generated by dysfunctional mitochondrial electron transport chain complexes to the p38 MAPK stress response pathway; (c) the pathway contributes to the sensitivity and vulnerability of aged tissues to diseases of oxidative stress; and (d) the importance of inhibitors of these pathways to the development of cardioprotection and pharmaceutical interventions. We propose that the ASK1-signalosome regulates the progression of cardiovascular diseases. The resultant attenuation of the physiological characteristics of cardiomyopathies and aging by inhibition of the ASK1-signalosome network lends support to this conclusion. Importantly the ROS-mediated activation of the ASK1-signalosome p38 MAPK pathway suggests it is a major center of dissemination of the ROS signals that promote senescence, aging and cardiovascular diseases. Pharmacological intervention is, therefore, feasible through the continued identification of potent, non-toxic small molecule inhibitors of either ASK1 or p38 MAPK activity. This is a fruitful future approach to the attenuation of physiological aspects of mammalian cardiomyopathies and aging.

## 1. Introduction

The complex physiological signal transduction networks that respond to inflammatory and and/or oxidative stress (ROS) challenges are major factors that promote the expression of senescence characteristics (in vitro-cells in culture) and aging (in vivo, animal and human models). These factors play a key role in the development of cardiovascular pathologies. Furthermore, these signaling networks contribute to the development of age-associated diseases of oxidative stress, that suggest cross talk between challenges of inflammation and oxidative stress and the development of senescence, aging and cardiovascular disease CVD. Furthermore, the inhibition (or attenuation/suppression) of these signaling networks delays the onset of these diseases. It appears, therefore, that these inflammation-ROS responsive signaling networks encompass the physiological processes of (a) senescence and aging, (b) oxidative stress and inflammation and, (c) cardiomyopathy (diseases of inflammation and oxidative stress) and in healthspan and lifespan determination. In this review, we discuss the molecular mechanisms associated with these physiological responses to inflammation and oxidative stress and emphasize the nature of the crosstalk by these signaling processes.

Inflammation and oxidative stress are major sources of both endogenous, e.g., sterile inflammation [1] and exogenous challenges that promote senescence and aging phenotypes. There is, therefore a physiological interaction that links inflammatory and ROS processes, to the activation of downstream networks that promote the physiological characteristics of senescence, aging, and cardiomyopathies. We will address the significance of the cardio-protection caused by attenuating these pathways, and the role of these inhibitors in the regulation of healthspan and lifespan extension. The inflammation-oxidative stress cycle of physiological interactions that promote senescence and aging, may thus favor the progression of such diseases of oxidative stress, as cardiomyopathies. Our hypothesis proposes that inflammatory and ROS challenges activate common signaling processes whose integration targets the downstream promotion of senescence, aging and CVD (Figure 1).

A major pathophysiological characteristic of aging involves the elevated and sustained endogenous level of expression of the sterile stress response signaling pathways that involve p38 MAPK, SAPK/JNK and NFkB. These are the major signaling pathways [1,2,3] that promote the initiation and progression of senescence and CVD phenotypes. There is, therefore, an elevated and sustained state of chronic inflammatory-oxidative stress that promotes the age-associated state of chronic stress and may thus play a key role in conferring increased age-associated tissue vulnerability to initiation and progression of these diseases [1,2,3] The signaling networks of tissues in an age-associated state of chronic stress are thus indicative of inflammatory processes that promote elevated endogenous inflammation and oxidative stress derived from senescence activated secretory protein (SASP) inflammation. These physiological signaling processes are key factors that promote the development of characteristics of stress-induced-aging and declining tissue functions of aging. These characteristics also contribute to the progression of age-associated diseases of oxidative stress.

Oxidative stress is a major causative factor of many age-associated cardiac disorders that include ischemia/reperfusion, hypertensive heart disease, diabetes, etc. These cardiovascular disorders exhibit endogenous levels of inflammation-promoting factors, i.e., cytokines. The induced/increased oxidative stress (ROS)-responsive signaling pathways are targeted by inflammatory-oxidative stress and ROS-generating systems, such as NADPH-oxidases and misfolded protein stress responses of the endoplasmic reticulum and mitochondria (Figure 1 and Figure 2).

The significance of inflammation-ROS activated physiological signaling networks to cardiovascular disorders is evident from studies demonstrating that oxidative stress may play a role in the declining cardiac contractility of sarcomeres due to oxidative mediated post-translational modification (PTM) of myofilament and titin proteins [4,5]. These studies emphasize the complexity of cellular responses to inflammation-ROS syndrome and that both induced (environmental challenges) and endogenous ROS (age-associated ETC dysfunction) are causative factors that promote the progressive age-associated declines in tissue functions [6,7,8,9]. Thus, we have proposed that the stress response signaling pathways whose activities are elevated and stabilized in response to age-associated endogenous and induced inflammation-ROS physiology in aged tissues play a key role in tissue vulnerability i.e., lowered resistance to the initiation of diseases of oxidative stress. These inflammation-associated signal transduction processes serve as centers of dissemination that promote senescence, aging and diseases of oxidative stress. Reactive oxidative radicals of endogenous as well as environmental (exogenous) origin enhance these signaling processes thus linking, for example, exposure to toxicants with accelerated aging and promotion of disease. Exposure to elevated levels of the chronic inflammation-oxidative stress thus regulates signaling pathways that control the development of aging and/or susceptibility to diseases of oxidative stress. The suppression of these pathways delays the accumulation of senescence products that promote extended healthspan and lifespan phenotypes.

Mitochondrial ETC dysfunction have been identified as a major source of age-associated ROS [10,11,12,13] thus emphasizing the importance of understanding the mechanisms that link mitochondrial dysfunction (ROS) to ROS-responsive signaling pathways that promote the characteristics of senescence, aging and diseases of oxidative stress.

## 2. The ASK1-Signalosome, a ROS-Responsive Signaling Complex (Hub), Links Mitochondrial Generated ROS to Senescence, Aging and Age-Associated Cardiovascular Diseases

The physiological responses to endogenous and/or environmentally induced ROS generated by mitochondrial ETC dysfunction involves a complex series of ROS-sensitive protein-protein interactions, protein conformational changes and protein modifications that result in the formation of macromolecular complexes that promote physiological processes of senescence, aging and age-associated diseases. The physiological responses to these conformational changes involve the assembly and interaction of stress response signaling networks that include the activation of the p38 MAPK, SAPK/JNK and NFκB pathways [14,15,16]. These pathways play a major role in the promotion of senescence and aging, and importantly they promote age-associated diseases of oxidative stress. In fact, there is ample evidence that transduction of mitochondrial generated ROS signals due to ETC dysfunction is mediated via the ASK1-signalosome→p38 MAPK/SAPK-JNK pathways [17,18,19,20]. This raises the question of the mechanism that links the ROS signals to these pathways; the nature of ROS-responsive signaling protein complexes; the mechanisms that target the networks that activate the physiological characteristics of senescence, aging and age-associated diseases of oxidative stress.

## 3. The Structure, Function and Mechanism of Regulation of the ASK1-Signalosome

The ASK1-signalosome is a high molecular weight (HMW) protein complex (~1500 kDa) composed of ROS-sensitive inhibitor and activator proteins whose overall functions regulate the response to ROS and the signaling networks that promote senescence, aging and age- associated diseases of oxidative stress (Figure 3). The complex serves as a center of distribution of inflammation and ROS signals [17,20,21] that activate or repress major stress response pathways that include cardiopathology, i.e., the p38 MAPK, SAPK/JNK and NFκB [14,15,16] pathways. Its unique protein components function as potential transducers of inflammatory tumor necrosis factor α and bacterial lipopolysaccharide (TNFα, LPS) and mitochondrial ETC dysfunction (ROS signals) that target the physiological processes of senescence, aging and diseases of oxidative stress including cardiomyopathy (Figure 1). The maps of complexed ASK1 (Figure 3) show the critical binding domains for regulatory proteins that form the ROS-responsive ASK1-signalosome which consists of an inhibitory N-terminal domain that binds reduced thioredoxin [(SH)_2_Trx], a catalytic kinase domain, and a C-terminal regulatory domain that activates ROS-sensitive activator proteins that make up the activated ASK1-signalosome.

## 4. Models of the Assembly of the Inhibitory and Activated ASK1-Signalosome

There are two models for the mechanism of assembly of the ASK1-signalosome (Figure 4).

**Model 1**: The (SH)_2_Trx-ASK1 complex serves as an attenuator of the ASK1-signalosome→p38 MAPK-JNK pathways. Association of the (SH)_2_ Trx with ASK1 maintains the ASK1-signalosome in an inactive form (Figure 3); ROS-mediated oxidation of ASK1-bound (SH)_2_Trx stimulates dissociation of the Trx from its N-terminal binding site thereby enabling activation of ASK1 (in its dimerized form) and its downstream targets [22,23,24,25,26,27].

**Model 2**: In the ASK1-oxidation model, ROS (H_2_O_2_) induces disulfide bond formation between ASK1 monomers to form dimers (Figure 3). Both dimerization and phosphorylation are required for ASK1 activation. The role of (SH)_2_Trx is to reduce the disulfide bonds of the dimers which decreases their activity by monomerization. This model is based on the H2O2-mediated activation of ASK1 signaling [28,29,30,31,32]. The ASK1-signalosome thus regulates the levels of expression of its downstream targets, MKK3/6, and p38 MAPK activation [2,19]; MKK4/7 and JNK activation [33,34] (Figure 2 and Figure 6). The mechanism of activation of the p38 MAPK and SAPK/JNK, in response to mitochondrial generated ROS involves the dissociation of the reduced thioredoxin-ASK1 complex [(SH)2 Trx-ASK1] [17,19,22,23,33]. By this mechanism, the inhibitory form of the ASK1-signalosome serves as a negative regulator of the ASK1→p38 MAPK/SAPK•JNK pathway [17,23,33]. The inhibitory ASK1-signalosome exists as an inactive-cytoplasmic high molecular weight complex that is 14-3-3 sequestered in unstressed cells. In this mechanism the ROS-mediated oxidation of ASK1-bound Trx(SH)2 stimulates release from the 14-3-3 complex. This enables ASK1 dimerization and the assembly of activating proteins at the *C*-terminal end Figure 4 and Figure 5) [23,26,35,36].

## 5. The ROS-Sensitive Regulatory Proteins of the ASK1-Signalosome

ASK1 is composed of three regulatory C-terminal domains (Figure 4). These are (a) 14-3-3, an inhibitory sequestering docking site [37]; (b) AIP1, the ASK1 **i**nteracting **p**rotein that responds to ROS by facilitating the dissociation of ASK1-ASK2 heterodimer from its 14-3-3 inhibitor [38], and (c) **h**omodomain **i**nteracting protein kinase 1 (HIPK1), a nuclear sumoylated protein that is desumoylated in response to ROS and translocated to the cytoplasm where it associates with AIP1-ASK1 and induces the release of Trx and 14-3-3 from ASK1 [39]. These regulatory proteins form the ROS-sensitive ASK1-signalosome which is a ROS sensory center that distributes signals of oxidative stress to the p38 MAPK and SAPK/JNK pathways thereby promoting senescence, aging and age-associated diseases of oxidative stress.

The (SH)2Trx-ASK1 complex in the inhibitory form of the ASK1-signalosome is dissociated by mitochondrial ETC-generated ROS, i.e., rotenone (ROT) an inhibitor of complex I (CI); 3-nitropropionic acid (3-NPA) an inhibitor of complex II (CII) and antimycin A (AA), an inhibitor of complex III (CIII) [17,19]. This mechanism links mitochondrial generated ROS to the activation of the ASK1-signalosome→p38 MAPK [17,19] via its downstream substrates, MKK3 and MKK6, and the ASK1-signalosome→SAPK/JNK via MKK4 and MKK7 (Figure 5).

Linking mitochondrial-generated ROS to the activation of p38 MAPK presents a potential mechanism for the ROS-mediated activation of pathways that promote characteristics of aging and senescence (p16Ink4a and p19Arf) via p38 MAPK [40] (Figure 6). This mechanism also identifies the physiological processes that sustain elevated p38 MAPK activity in aged tissues thereby promoting this characteristic of aging [19]. The sustained elevated p38 MAPK activity also occurs in myocardial ischemia and may thus be a key physiological factor in the progression of the myocardial ischemia phenotype [41].

## 6. The p38 MAPK and SAPK/JNK Pathways

Our studies have described a mechanism that links mitochondrial-generated ROS to the activation of stress induced aging phenotypes and that the ROS-responsive ASK1-signalosome regulates the p38 MAPK pathway activity and its downstream targets of senescence and aging, e.g., p16^Ink4a^, p19*^Arf^*, TNFα, JNK, apoptosis and cardiovascular diseases (Figure 1 and Figure 6). These studies suggest that development of certain diseases of oxidative stress are linked to the ASK1→p38 MAPK pathway [19] for example, hypertension induces the cell cycle inhibitor, p16^Ink4a^ which is a characteristic of somatic cell senescence and is activated by p38 MAPK in rat and human endothelial cells [42]. We have thus proposed that the ASK1-signalosome distributes signals of oxidative stress and inflammation by activating p38 MAPK which targets the pathways of aging, senescence and age-associated cardiovascular diseases Figure 6). The fact that ~90% of age-associated ROS originates from mitochondrial dysfunction and that the p38 MAPK and SAPK/JNK pathways are activated by mitochondrial generated ROS strongly supports the hypothesis that these signaling pathways promote characteristics of senescence, aging and diseases of oxidative stress.

## 7. Oxidative Stress Generated by Mitochondrial Dysfunction Elevates and Sustains p38 MAPK Activity and Promotes Senescence, Aging and Cardiovascular Disease via the ASK1-Signalosome

Increased and sustained levels of p38 MAPK and SAPK/JNK activities are major physiological characteristics of aging [1,2,43,44]. Interestingly, these physiological characteristics occur in age- associated CVD. The elevated age-associated endogenous activity of many of the stress response genes targeted by p38 MAPK may thus be a consequence of the sustained elevated activities of these pathways. We propose, therefore, that the chronic elevated levels of endogenous ROS are physiological characteristics that contribute to the promotion of senescence and [42] aging and to the vulnerability of aged tissues to diseases of oxidative stress including cardiovascular myopathies [1,2,10,11,40,43,45,46]. The physiological environment caused by the elevated levels of oxidative stress is favorable for the maintenance of elevated levels of ASK1-signalosome, and p38 MAPK activity thereby promoting characteristics of senescence in vitro and the induction of aging and CVD in vivo. Thus, the consequences of sustained elevated ASK1-signalosome→p38 MAPK and SAPK/JNK pathways results in the chronic stimulation of the downstream signals of senescence and aging (Figure 6). We propose that this is a physiological characteristic that plays a key role in the development of (CVD).

## 8. The ASK1-Signalosome Links Mitochondrial ROS to Activation of p38 MAPK and SAPK/JNK Pathways and Promotion of Senescence, Aging and Cardiovascular Diseases

The elevated levels of endogenous oxidative stress in aged tissues may promote senescence signaling pathways targeted by p38 MAPK. Furthermore, the decreased levels of oxidative stress that occur in various long-lived mouse models may be a factor that delays senescence pathways by attenuating p38 MAPK activity [17,18,47,48].

The endogenous levels of the inhibitory ASK1-signalosome in resting, unchallenged cells are altered by ROS generated by mitochondrial ETC dysfunction (Figure 5 and Figure 6). Thus, the dissociation of the (SH)_2_Trx-ASK1 complex and association of regulatory proteins at the C-terminus activate the ASK1-signalosome and is part of the molecular mechanism that maintains the elevated endogenous p38 MAPK activity in aged mice [1,18,19,40,43,48,49]. This mechanism is supported by our studies with Snell and Ames dwarf long-lived mice in which we have shown that the endogenous level of the inhibitory ASK1-signalosome is significantly higher in these oxidative stress resistant long-lived models and in Ames dwarf fibroblasts in culture [17,19]. The lower p38 MAPK activity in these long-lived mice is consistent, therefore, with their resistance to oxidative stress. A similar correlation of ASK1-signalosome activity occurs in the oxidatively resistant Klotho overexpressing mouse model (elevated levels of the inhibitory signalosome) and in the oxidatively stressed Klotho(−/−) model (elevated levels of the activated signalosome) [18].

## 9. Promotion of Senescence and Aging

The sustained elevated level of p38 MAPK activity chronically stimulates the downstream signals of senescence and cardiovascular pathophysiology. Thus, expression of the Ink4a and Arf cell cycle inhibitors (of the Cdkn2a tumor suppressor locus) which are also promoters of senescence and aging [50], are chronically elevated and their sustained activity, i.e., p16^Ink4a^ and p19^Arf^, which increase markedly is associated with senescence and aging [51]. Recent studies have indicated that TNFα activation of the p38 MAPK→p16^Ink4a^ promotes the senescence of endothelial progenitor cells and demonstrates their vulnerability to atherosclerosis [52].

Several lines of evidence support a role for p38 MAPK in the regulation of mammalian senescence and aging via the expression of p16^Ink4a^ and p19^Arf^ [53]. The role of p38 MAPK in activating these cell cycle inhibitors has been convincingly demonstrated by the development of a dominant-negative allele (p38^AF/+^) in which the catalytic Tyr^182^ phosphorylation site was substituted with Phe thereby inactivating the p38 MAPK activity but not its synthesis [54]. The heterozygous p38 ^AF/+^ mice show a marked attenuation of p38 MAPK-dependent signaling and age-induced expression of the p16^Ink4a^ and p19^Arf^ cell cycle inhibitors in different organs as well as in mouse embryonic fibroblast cells in culture. Furthermore, aged p38AF/+ mice show enhanced proliferation and regeneration of pancreatic islet cells when compared to wild type littermates, an indication of the attenuation of p16Ink4a and p19Arf in this mutant. Additional support of this mechanism is provided by the demonstration that the reduction of expression of Wip1 phosphatase in aged mice, and loss of activity in Wip1 deficient mice, results in elevated levels of P-p38 MAPK activity and decreased islet proliferation [54,55]. On the other hand, Wip1 overexpression that attenuates p38 MAPK activity rescues the age-related decline in proliferation and regenerative capacity. Thus, the upregulation of P-p38, p^16Ink4a^ and p19^Arf^ expression demonstrates the role of P-p38 MAPK in the regulation of expression of cell cycle inhibitors and age-related decline of cell proliferation [55].

## 10. The Inhibitory ASK1-Signalosome and Resistance to Oxidative Stress. Thioredoxin, an Important Component of the Inhibitory ASK1-Signalosome: Its Role in Cardiovascular Disease

Thioredoxin-1 (Trx1) serves as an oxidoreductase that also interacts with other proteins such as the ASK1-signalosome via disulfide bridges. In some cases, it translocates to the nucleus where it binds to different transcription factors, i.e., regulates DNA-binding activity of p53, NFκB, and AP1. Trx2, another member of the family, is localized in the mitochondria, where it plays a role in cell growth and inhibition of apoptosis [56,57,58]. Both Trx1 and Trx2 have been implicated in cardiovascular disease in that they have cardioprotective activity [59].

Thioredoxin-1 (Trx-1) is a major ROS (redox) sensitive signaling protein whose ability to complex with other proteins results in the formation of protein complexes that control the signaling pathways that are (a) responsive to changes of oxidative stress; (b) activate signaling pathways associated with aging and age-associated CVD of oxidative stress.

The thioredoxin system plays a key role in the cardiovascular and smooth muscle cell physiology, e.g., in smooth muscle cell proliferation. Overexpression of Trx1 increases DNA synthesis in human aortic vascular smooth muscle cells [60]. Furthermore, growth factors such as PDGF, EGF and VitD3-upregulated protein 1(VDUP-1) increase oxidative stress and proliferation of smooth muscle cells and increase Trx1 activity in smooth muscle cells [60,61,62].

## 11. Resistance to Oxidative Stress

The levels of the (SH)_2_Trx-ASK1 form of the inhibitory ASK1-signalosome decrease while activated ASK1-signalosome levels increase in response to changes in mitochondrial generated ROS suggesting that this may be a physiological characteristic of tissue resistance to oxidative stress [18,19,47]. The mechanism of this resistance may thus involve a balance between the inhibitory ASK1-signalosome vs. the activated ASK1-signalosome based on its mediation of the levels of activities of the p38 MAPK and SAPK/JNK pathways. The levels of activities of these pathways are indeed a basic difference between progressions of aging in wild-type vs. the delayed aging in long-lived mice [17,18].

Recent characterization of the ASK1-signalosome has provided further evidence of its multiple functions. In resting cells the high molecular weight signalosome is bound to 14-3-3, and contains thioredoxin and MKK6 (Figure 4) [21]. Resistance to oxidative stress of the long- lived Snell and Ames dwarf mouse mutants and the overexpressing Klotho model is associated with their increased levels of (SH)_2_Trx-ASK1 complex [17,18,19]. Thus, a decrease of activated ASK1 accounts for the decreased activity of downstream targets and may be indicative of their resistance to oxidative stress. This occurs in the Snell and Ames long-lived mice suggesting a sustained lower level of activity associated with the lower level of endogenous oxidative stress in both young and aged dwarf mice [17]. Furthermore, the fact that the level of reduced thioredoxin [Trx(SH)_2_] is significantly higher in dwarf cells is consistent with the higher inhibitory ASK1-signalosome complex levels, lower stress signaling activity and resistance to oxidative stress.

## 12. The Role of the ASK1-Signalosome in Cardiovascular Disease

Activation of ASK1 occurs in cardiovascular injury caused by high-salt diet [63] rennin-angiotensin II [64,65,66]-aldosterone [63]; vascular remodeling [65], high-fat diet-induced insulin resistance and vascular endothelial dysfunction [67] and angiogenesis [68]. These studies y suggest that the ASK1-signalosome→p38 MAPK pathway may play a role in the progression of these cardiomyopathies.

**Cardiovascular injury.** Cardiovascular injury is closely associated with high-salt diet. The mechanism of its progression may involve the activation of the ASK1-signalosome network [63]. The major finding that ASK1 deficiency abolishes high-salt induced inflammation as well as characteristics of cardiovascular injury suggest that ASK1-signalosome may function as a center of distribution of signals that promote physiological CVD symptoms. Chronic salt loading enhances cardiac inflammation, fibrosis and vascular endothelial impairment, and is accompanied by activation of the ASK1-signalosome→p38 MAPK pathway. These symptoms suggest that the p38 MAPK targets drive these cardiovascular syndromes (Figure 5 and Figure 6). This hypothesis is strongly supported by the observation that ASK^(−/−)^ mice fed on a high-salt diet do not develop the pathophysiological characteristics of cardiovascular disease which include enhanced TGFβ-1, interstitial fibrosis, coronary perivascular fibrosis and inflammatory cell infiltration. Other symptoms that develop in ASK1^(−/−)^ mice include impairment of vascular endothelium-dependent relaxation by acetylcholine, increased vascular superoxide and Nox2.

**Ischemic Heart Disease.** The multiple stress signaling networks activated in response to myocardial infarction (MI) include the p38 MAPK pathway, which suggests that the senescence and aging pathophysiology is part of the MI phenotype. Ischemia activates the p38 MAPK pathway in cardiac cells and the role of p38 MAPK in ischemic cardiomyopathy is indicated by the improvement of cardiac function after inhibition of p38 MAPK Hypertension, cardiovascular diseases and stroke are all associated with high-salt intake [68,69,70,71]. It is likely, therefore, that high-salt intake promotes the ASK1-siganlosome→p38 MAPK pathway thereby promoting the physiological networks that are characteristic of cardiovascular injury [63].

**Mycardial Infarction.** Ischemic and oxidative stress, and activation of local and systemic hormonal systems such as rennin angiotensin-aldosterone, endothelin and sympathetic nervous system are multiple physiological stress stimuli that are activated in response to MI [72]. The p38 MAPK and SAPK/JNK pathways are major stress/inflammatory response pathways that are activated in cardiac cells in response to MI. These pathways, which are linked to the ASK1-signalosome, are activated in cardiac cells by multiple extracellular stimuli that include: ischemia [73]; hemodynamic stress [74]; and neurohormonal factors such as angiotensin II (AngII) [75,76] hypertrophy [77]; inflammation [75,78]; fibroblast proliferation [75]; and myocyte apoptosis [79]. There are, therefore, multiple p38 MAPK and JNK dependent processes that characterize the post-MI pathological symptoms. Thus, the activation of p38 MAPK/JNK may contribute to progressive left ventricle (LV) remodeling post-MI and to the transition to heart failure. Indeed, sustained p38 MAPK activation in the heart is associated with LV remodeling and dysfunction arising from various etiologies both in humans [80] and in animals [81,82]. Interestingly, by inducing or sustaining p38 MAPK activity the MI etiologic agents also activate the senescence and aging phenotypes. This suggests that the down regulation or inhibition of p38 MAPK activity which is cardioprotective may also delay characteristics of senescence and aging and promote longevity [83].

**Cardiovascular Fibrosis.** A major impact of MI involves the loss of functional myocardium at the site of injury. This challenge triggers LV remodeling characterized by necrosis and thinning of the infarcted myocardium, LV chamber dilation, fibrosis at the site of infarct and non-infarcted myocardium and hypertrophy of viable myocytes [72]. TGFβ1 is a major factor that stimulates tissue fibrosis [84,85], and plays a role in induction of cardiofibrosis. The role of ASK1 in this process is supported by the observations that TGF 1 expression is not increased in ASK^(−/−)^ mice and that they show less cardiac fibrosis.

Oxidative stress is a causative factor in cardiac inflammation and fibrosis, and vascular endothelial dysfunction [61,62,86,87]. Indeed, a high-salt diet significantly enhances superoxide in WT mice but not in ASK1-deficient mice. Furthermore, the increased cardiac superoxide is associated with enhanced Nox2, a subunit of NAD(P)H-oxidase, although the enzyme activities are not altered. There is however a decrease in Nox2 activity in high-salt fed ASK1^(−/−)^ mice which is attributed to the decreased level of oxidative stress in these ASK1-deficient mice. These observations indicate that enhancement of Nox2 by high-salt diet induces vascular endothelial dysfunction and cardiac injury and that these symptoms are attenuated in the absence of ASK1, and presumably the ASK1-signalosome→p38 MAPK pathway. In addition, in MI and diabetic cardiomyopathy, cardiomyocytes compensate for heart tissue damage via cardiac hypertrophy which involves myocyte gene reprogramming and the accumulation of extracellular matrix (ECM) proteins which play critical roles in ventricular fibrosis and remodeling [88,89].

**Neurohormonal Systems—Rennin-Angiotensin.** The rennin-angiotensin system (Angiotensin II) plays a key role in cardiovascular disease [88]. Furthermore, in cardiac myocytes, ASK1 is activated by AngII via angiotensin II type I receptor in response to oxidative stress and is involved in the induction of cardiac hypertrophy. This response to oxidative stress is mediated in part by activation of p38 MAPK and SAPK/JNK pathways via the activation of the ASK1-signalosome [65,90]. Suppression of AngII-mediated cardiac hypertrophy in ASK1-deficient mice thus suggests that the ASK1-signalosome p38 MAPK/SAPK-JNK play a role in AngII signaling and links the processes of oxidative stress to cardiac hypertrophy [64,67].

The impairment of vascular endothelial function by diet-induced diabetes also occurs in ASK1 deficient mice and is attributed to the attenuation of eNOS dimer formation and subsequent reduction of superoxide production. Thus, ASK1 activation in diabetic mice by AngII-dependent signaling is associated with superoxide accumulation and the induction of vascular endothelial cell dysfunction and remodeling. Importantly inhibition of ASK1 by a dominant negative ASK1ΔKR expression vector suppresses ASK1-signalosome activation thereby preventing cardiomyocyte apoptosis and heart failure progression even after the onset of hereditary cardiomyopathy [91].

## 13. Cardiomyopathies, Senescence, Aging and Longevity

Insulin-like growth factor-1 (IGF-1) and Sirtuin (Sirt-1) are important mediators of cell survival, oxidative stress, regeneration and life span and play a key role in cardioprotection against oxidative stress (Figure 7). The onset of age-associated diseases including cardiovascular diseases is delayed in mice by caloric restriction and by SIRT-1, leading to a prolonged life span [91]. SIRT-1 increases upon calorie restriction in several rodents and human tissues, e.g., white adipose, liver, skeletal muscle, brain and kidney thus suggesting that it plays a role in longevity determination.

SIRT-1 is an NAD+-dependent deacetylase; the deacetylation reaction removes acyl groups from lysine side chains of a protein substrate while cleaving NAD^+^ in the process to generate the deacetylated protein 2′-*O*-acetylADP-ribose and nicotinamide (Figure 8). SIRT-1 influences many physiological processes that involve senescence, stress resistance, gene expression apoptosis and energy balance (Figure 9). Some of the longevity-associated characteristics of SIRT-1 involve the activation of PGC1α by deacetylation of lysine residues that results in increased mitochondriogenesis. A decline in mitochondrial function with age may be a major factor that contributes to the development of cardiovascular disease [92,93].

**IGF-1-Sirtuin1-Cardiovascular Disease.** The physiological signaling processes that extend life span, i.e., the IGF-1→SIRT-1 pathway, also protects against oxidative stress generated by AngII and paraquat [94]. The pathways in Figure 8 show the interactions of the IGF1→SIRT-1 pathway associated with extended life span and protection against ROS-mediated hypertrophy of cardiomyocytes. Thus, SIRT-1 exerts its protective effects by its anti-inflammatory functions which include cardioprotection against oxidative stress, both endogenous by AngII and induced by paraquat, and in general, the aging of the heart [95]. Its overexpression in mice exhibits some physiological characteristics of calorie-restricted (CR) mice [95,96]; its beneficial effects are seen in Alzheimer’s and Huntington’s disease models [97,98] and include the activation of PGC1 and its involvement in mitochondrial biogenesis [94,99]. In addition to its cardiomyocyte protective effects and anti-senescence and aging properties, the complexity of its activity is indicted by its multiorgan protective physiological processes that ameliorate/delay age-associated diseases many of which are diseases of oxidative stress (Figure 8 and Figure 9; [94,99]). Furthermore, genetic evidence of its cardioprotective role is indicated by single nucleotide polymorphism (SNP) in the human SIRT-1 gene, which results in lower incidence of cardiovascular mortality, myocardial infarction, myocardial ischemia, stroke, arterial surgery and intermittent claudication [100].

## 14. Atherosclerosis-Premature Senescence of Vascular Endothelial Cells

Vascular endothelial cells also exhibit the characteristics of senescence and aging that are promoted by oxidative stress and inflammation [52]. This suggests a physiological link between systemic aging and atherosclerosis Analysis of the senescence characteristics of the highly proliferative epithelial progenitor cells (EPCs) and their vulnerability to inflammatory stress [52] has shown that these EPCs have (a) very low basal levels of the physiological characteristics of senescence compared to mature ECs; (b) that p16^Ink4a^ is expressed at higher levels in the differentiated ECs than in proliferative EPCs; (c) that exposure to chronic TNFα up-regulates expression of p38 MAPK and the senescence associated p16^Ink4a^, both of which play a role in cell cycle arrest, and (d) that inhibition of p38 MAPK blocks the induction of p16^Ink4a^ and cellular senescence [101]. These studies demonstrated that highly proliferative EPCs have low levels of intrinsic, baseline senescence compared to mature ECs and are therefore, vulnerable to the induction of premature senescence by chronic proinflammatory stimulation.

The induction of p16^Ink4a^ and TNFα-mediated chronic inflammation and oxidative stress suggests that these factors may be involved in the mechanism of the loss of embryonic progenitor cell (EPC) proliferative capacity and their increased vulnerability to aging and atherosclerosis [102,103]. Chronic inflammatory stimulation is, therefore, a significant in vivo physiological environment that promotes premature EPC senescence and atherosclerosis. This physiological environment may also account for the loss of self-renewal due to the loss of proliferative capacity, a phenomenon that exhausts the repair potential of the blood vessels. Furthermore, the loss of repair may potentially play a role in conferring higher EPC vulnerability to disease development. We thus propose that sustained elevated expression of p38 MAPK due to endogenous chronic sterile inflammation and oxidative stress drives the EPCs out of the cell cycle thereby decreasing the stem cell population which is known to occur in aged tissues. Further support for a role of cell cycle inhibitor-senescence associated genes is provided by the fact that one of the key genetic loci identified as a risk factor for MI is located on chromosome 9p21, in an area that encodes p16Ink4a [104]. Furthermore, there is evidence that p38 MAPK→p16^Ink4a^ activates senescence in other stem cell types [103] and that they are also highly vulnerable to stress-induced premature senescence. The increased vulnerability of age-associated regenerative cardiovascular cells to oxidative stress is a long-standing question whose mechanism may indeed involve the balance of activity of the ROS-responsive and sensitive ASK1-signalosome→p38 MAPK and its downstream-targeted pathways. Thus, with respect to vulnerability to diseases of oxidative stress, we propose that the increased levels of age-associated endogenous oxidative stress and chronic inflammation elevates and sustains the levels of the ASK1-signalosome→p38 MAPK pathway thereby priming the promotion of senescence, aging and proathersclerotic pathophysiological characteristics in aged tissues.

Activation of p38α MAPK occurs during remodeling of damaged cardiac tissue, i.e., after MI. Atherosclerotic lesions which are the cause of many forms of cardiovascular disease are characterized by lipid-laden macrophage foam cells that arise from p38MAPK dependent uptake of oxidized low density lipoprotein [105]. The mechanism of activation and/or sustained activity of p38 MAPK in these pathologies may be linked to the ROS-mediated activation of the ASK1-signalosome→p38 MAPK pathway.

## 15. Oxidative Stress (p66Shc), Senescence, Aging and Cardiovascular Disease

Oxidative stress plays a key role in the signaling networks that regulate vascular homeostasis, myocardial and vascular disease and aging [101,105]. Although ROS are generated at several intracellular sites, mitochondria are an established principal source of ROS. The protein p66Shc, an isoform of the growth factor adaptor, Shc, plays an important role in the generation of mitochondrial ROS. Ablation of the gene extends life span by about 30% with no pathological consequences [106]. The mutant also displays increased resistance to oxidative stress and decreased levels of intracellular ROS whereas in the overexpressing models the increased generation of mitochondrial ROS is a triggering mechanism for the development of CVD [101]. Thus, the deletion of p66Shc confers protection against diabetes- related (high glucose levels) cardiovascular symptoms. This mutant has thus provided strong evidence for the role of ROS generated by mitochondrial dysfunction in senescence and aging [106,107,108,109,110].

The role of p66^Shc^ in generating mitochondrial ROS involves its transport into the intermitochondrial space (IMS), which is accomplished by the binding of p66Shc with mitochondrial HSP70, a TOM-TIM complex. Release of p66Shc into the inter mitochondrial space IMS (IMS) results in its activation and oxidation of cytochrome c thus stimulating the production of H_2_O_2_ at ETC complex IV, and initiation of apoptosis. The mechanism of activation of p66Shc in the IMS is not known, although its transport from the cytoplasm involves the free radical activation of protein kinase C-β, which phosphorylates Ser^36^ thereby activating its transfer to the IMS.

The physiological basis for proposing a role of p66Shc in cardiovascular disease lies in its localization to the IMS where it acts as an ROS generating oxidoreductase leading to mitochondrial dysfunction and cell death [108,111,112]. The p66^Shc^ system thus provides an alternative redox process associated with mitochondrial ETC complex IV dysfunction that generates proapoptotic ROS in response to stress signals. The p66^Shc^ system thus provides a genetic mechanism that links ROS generated by mitochondrial dysfunction to the pathophysiology of cardiovascular risk factors.

It has been proposed that p66Shc signaling promotes the pathophysiological characteristics of high-glucose associated endothelial dysfunction and cardiomyopathy. Thus, p66^Shc^ links the ROS generated by mitochondrial dysfunction at ETC complex IV to the promotion of physiological characteristics of senescence and aging based on its role in the determination of extended life span, increased resistance to oxidative stress, and physiological characteristics of cardiovascular disease.

## 16. Therapeutic Applications to the ASK1-Signalosome and p38 MAPK

The severe inflammatory responses to myocardial dysfunction (myocardial infarction) are initiated in the myocardium, causing excessive chronic elevation of ROS and inflammatory cytokines [113,114,115,116]. The design of anti-inflammatory drugs with emphasis on treatment of CVDs has focused on the inhibition of p38 MAPK induced inflammation due to enhancement of cytokines.

The significance of p38 MAPK in promoting CVD as well as the senescence and aging phenotypes is indicated by the demonstration that selective inhibitors of p38α and p38β such as RWJ-67657 (RWJ) and SB203580 attenuate their progression [117,118,119,120,121]. Inhibition of the p38 MAPK cascade by treatment of rats after MI with RWJ attenuates the natural progression of pathologic LV remodeling and dysfunction [121]. These results indicate that p38 MAPK signaling targets the pathways that promote pathologic remodeling after MI as well as the progression of physiological characteristics of senescence and aging. Genetic evidence of this physiological p38 MAPK function is provided by delay of the appearance of senescence markers in the dominant negative p38^(AF/−)^ mouse [122]. Furthermore, a role for p38 MAPK in promoting cardiomyopathies is suggested by the observation that long-term pharmacological blockage of p38 MAPK reduces hypertrophy and dysfunction and enhances survival of spontaneously hypertensive stroke-prone rats maintained on high-salt/high fat diet [82]. Inhibitors of p38 MAPK also improve myocardial function of failing human heart owing to ischemic injury [123]. Together, these data indicate that p38 MAPK pathway plays an important role in the promotion of LV remodeling and dysfunction and heart failure disease progression. We propose that the ASK1-signalosome→p38 MAPK pathway enhances the progression of oxidative stress promoted myocardial dysfunction in cardiomyopathies [124].

The cardioprotective effect of p38 MAPK inhibition by SB203580 is attributed to down regulation of proinflammatory TNFα thereby attenuating LV remodeling [125]. The overall therapeutic response to the cardioprotection of SB203580 includes reduced progression of myocardial fibrosis, decreased TNFα production and down regulation of collagen Type I all of which are physiological markers whose expressions are associated with the progression of LV remodeling [126], and in the promotion of senescence and aging phenotypes. The fact that TNFα is induced by p38 MAPK and that elevated levels of TNFα sustain p38 MAPK activity suggests a locked-in signaling cycle in which TNFα generated ROS activates the ASK1-signalosome→p38 MAPK pathway and its downstream target, TNFα. The inhibition of p38 MAPK attenuates TNFα secretion and reverses myocardial fibrosis leading to improved cardiac function [73,82]. Interestingly, TNFα is also an activator of mitochondrial apoptosis which is attributed to the activation of the ASK1-signalosme→SAPK/JNK apoptosis pathway [23]. Similarly, the suppression of p38 MAPK activity in vivo in the p38^(AF/+)^ heterozygote and in cells derived from this mutant suppresses the activation of senescence and aging markers [122]. These studies suggest that the ASK1-signalosome p38 MAPK is a common signaling pathway that targets senescence, aging and cardiomyopathy. A major activity of p38 MAPK is its inhibition of cell cycle check points [127] resulting in growth arrest, apoptosis [53,128,129] and cellular senescence [52,130].

## 17. Other Diseases of Oxidative Stress

The regulation of ASK1 is implicated in various diseases of oxidative stress, including neuronal cell death and ROS production in neurodegenerative diseases such as Parkinson’s [44,131,132], Alzheimer’s [133], and amyotrophic lateral sclerosis [134]. It is interesting that the incidence and prevalence of Parkinson’s disease is higher in males vs. females [135] and the neuroprotective action of estrogen is acknowledged to be the underlying mechanism of this protection [131,136]. This gender difference, seen in human populations, is reflected in the mouse model wherein female mice demonstrate an innate protection from MPTP (1-methyl-4-phenyl-1,2,3,6- tetrahydropyridine)-mediated neurotoxicity [131,132]. This is attributed to the estrogen-mediated activation of thioredoxin reductase, which plays a critical role in establishing elevated levels of the (SH)_2_Trx-ASK1 form of the inhibitory ASK1-signalosome. Thus, the levels of activity of redox activated p38 MAPK signaling cascade(s) that are implicated in the selective degeneration of SNpc dopaminergic neurons in mice are significantly increased [132,137]. The mitochondrial dysfunction at ETC complex I caused by MPTP is absent in females and is, thus, consistent with the proposed mechanism that ROS generated by complex I dysfunction activates the p38 MAPK pathway via dissociation of the inhibitory ASK1-signalosome [17,18,19]. The higher levels of the inhibitory (SH)_2_Trx-ASK1 form of the signalosome is thus maintained by increased levels of thioredoxin reductase activity which attenuates p38 MAPK activity in females (Figure 1) (Figure 5 and Figure 6) This is consistent with the cardioprotective functions of p38 MAPK inhibitors discussed above.

p38 MAPK plays an important role in induction of cellular senescence by a diverse set of stimuli, e.g., Ras-induced senescence which is due to an alternative source of oxidative stress [44,138,139]. The small GTPases, are another source of endogenous ROS that activate p38 MAPK [140]. Future studies will assess whether the small GTPase-generated ROS enhancement of senescence and aging is mediated via the ASK1-signalosome. In addition, several studies have identified ETC proteins that are oxidatively modified in aged tissues. These modified proteins are potential generators of ROS via the misfolded protein stress response and raise the question of whether misfolded protein-generated ROS contribute to the promotion of senescence, aging and cardiomyopathies, and does this involve the activation of the ASK1-signalosome→p38 MAPK →senescence pathways?

## 18. The Oxidative Stress-Mediated Post-Translational Modification of Cardiac Muscle Proteins—Post-Translational Modification of Sarcomere Proteins

The oxidative stress-mediated post-translational modification (PTM) of proteins is a basic physiological factor associated with age-associated loss of protein functions as well as ROS generation by the misfolded protein stress response. Both physiological characteristics are attributed to increased levels of inflammation and oxidative stress. In this respect it is significant that miscommunication between sarcomeric proteins and contractile dysfunction is attributed to oxidative stress-mediated PTM of myofilament proteins [4,27].

The specificity of PTM is suggested by studies showing that ROS can preferentially act on regions of the myofilamental proteins exposed to Ca^2+^ activation as opposed to inaccessible regions of attached cross-bridges [141]. This ROS modification induces effects that disrupt the integrity of the sarcomeric lattice.

The ability of certain proteins to serve as redox sensors is a basic physiological function carried out by certain tissue specific proteins as well as ASK1. Myosin heavy chain (MHC) is a tissue specific protein that serves as a redox sensor in the sarcomere due to its redox modifications at Cys^697^ and Cys^707^ which decreases myosin ATPase activity and leads to myofilament dysfunction [4,142,143,144,145]. Other proteins that are ROS sensitive and subject to redox modifications include actin and tropomyosin, which result in defects in actin-myosin cross bridge formation and thin filament activation by Ca^2+^ [146]. This is attributed to oxidation of Cys^374^, which results in changes in actomyosin ATPase activity and actin filament sliding velocity [147].

When cardiomyocytes experience oxidative stress, disulfide bridges are formed in the titin N2-B_us_ domain, which is a physiologically extensible region capable of S-S bonding. This increased oxidative stress due to modification elevates titin-based stiffness of cardiomyocytes, which contribute to the global myocardial stiffening frequently seen in the aging or failing heart [5,148,149,150]. This disulfide bonding occurs specifically in titin’s N2-Bus region, thus showing an important differential modification associated with the ROS-mediated loss of function of this protein. Oxidative modifications such as S-S bonding in titin’s N2B_us_ may add to other aging- or disease- related mechanisms that modify titin-based stiffness, such as titin-isoform transition and alterations in titin phosphorylation [151]; ligand binding [146], mechanosensing [152], or phosphorylation-dependent stiffness adjustment [153], thus emphasizing the S-S bonding property of the N2-B_us_, which is likely to be of pathophysiological importance.

## 19. Conclusions

The ASK1-signalosome network is a major center of distribution of ROS-mediated stress signals that plays multiple roles in promotion of senescence, aging and diseases of oxidative stress one of which is cardiomyopathy. Its inhibition attenuates the development of these physiological diseases and its attenuation is associated with increased life span. Importantly the association of ASK1-signalosome→p38 MAPK oxidative stress suggests that p38 MAPK is a major center of dissemination of the ROS-activated ASK1-signalosome thus linking signaling networks that drive senescence, aging and diseases of oxidative stress. Thus, the ASK1-signalosome and p38 MAPK are potential master regulators of the activity of many stress response genes associated with aging. Our proposed mechanism implies that the ASK1-siganalosome linked regulation of p38 MAPK activity should have multiple gene targets associated with resistance and sensitivity to oxidative stress as well as diseases of oxidative stress.

Our mechanism suggests that promotion of the premature aging phenotypes as well as cardiomyopathies may be regulated by ASK1-signalosome→p38 MAPK activity. Thus, the use of inhibitors that attenuate this pathway, either at the level of ASK1, MKK3 or p38 MAPK could be beneficial in ameliorating such conditions. Although detailed studies have shown the cardioprotective effects of p38 MAPK inhibitors, recent advances in high throughput screening of small molecule libraries have identified benzodiazepine as a potent inhibitor of ASK1. Pharmacological intervention is, therefore, feasible through the continued identification of potent, non-toxic small molecule inhibitors of either ASK1 or p38 MAPK kinase activity. This is a fruitful future approach to the attenuation of physiological aspects of mammalian aging and cardiomyopathies.

## Figures and Tables

**Figure 1 cells-08-01383-f001:**
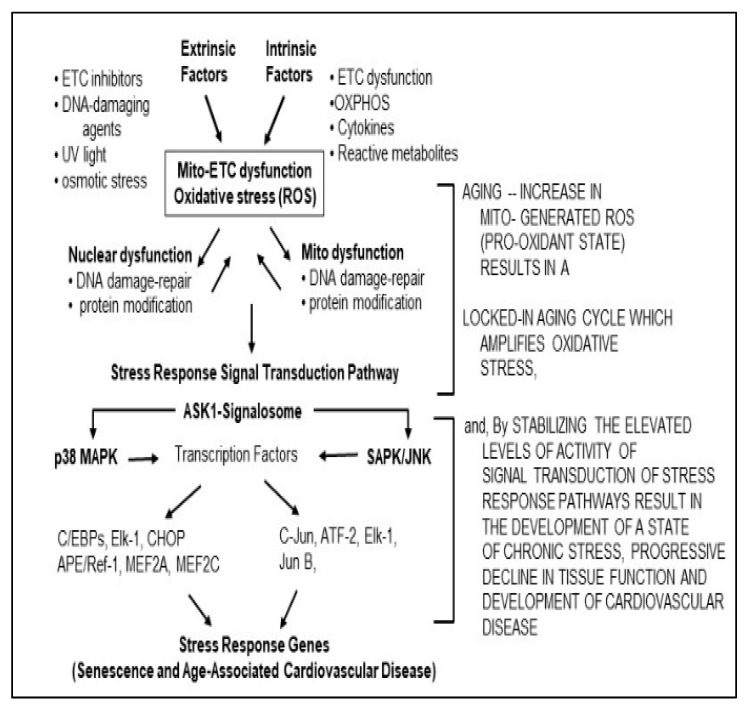
The oxidative stress chronic cycle of aging defines the epigenetic processes in development of the biochemical characteristics of senescence, aging and aging-associated diseases.

**Figure 2 cells-08-01383-f002:**
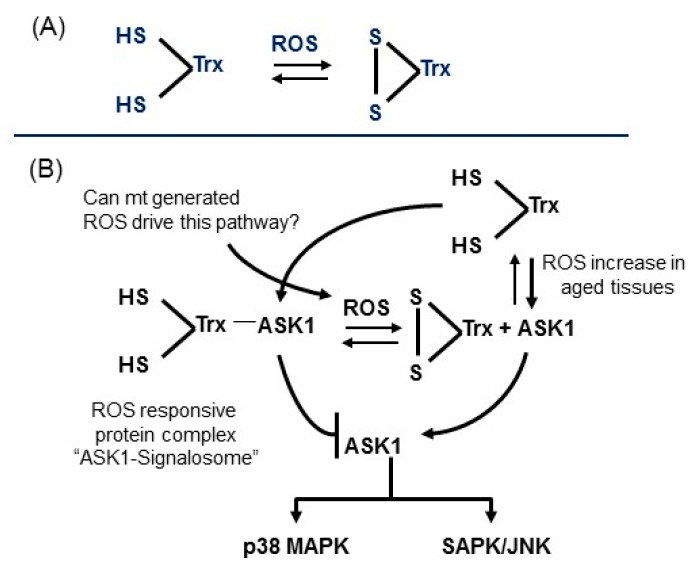
Thioredoxin links the production of ROS to the activation of stress response pathways. (**A**) Thioredoxin plays a major role in the regulation of the redox state of the cell. (**B**) The redox state of thioredoxin plays a key role in the regulation of the ASK1→P38 MAPK and SAPK/JNK stress response signaling pathways.

**Figure 3 cells-08-01383-f003:**
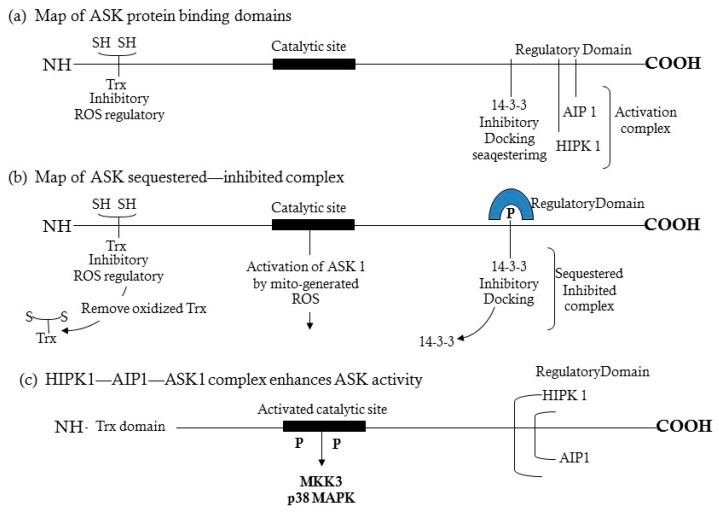
Maps of the protein binding domains of the ASK 1-signalosome and the protein-protein interactions regulated by ROS activation of the signalosome. (**a**) The reduced thioredoxin ASK1 complex which is part of the inhibitory ASK1-signalosome; (**b**) The inhibitory ASK1-signalosome is sequestered as a 14-3-3 complex; (**c**) The components of the activated ASK-signalosome that bind to the C-terminal end of ASK1. Modified from Li, X. et al. (2005) JBC 280 15061.

**Figure 4 cells-08-01383-f004:**
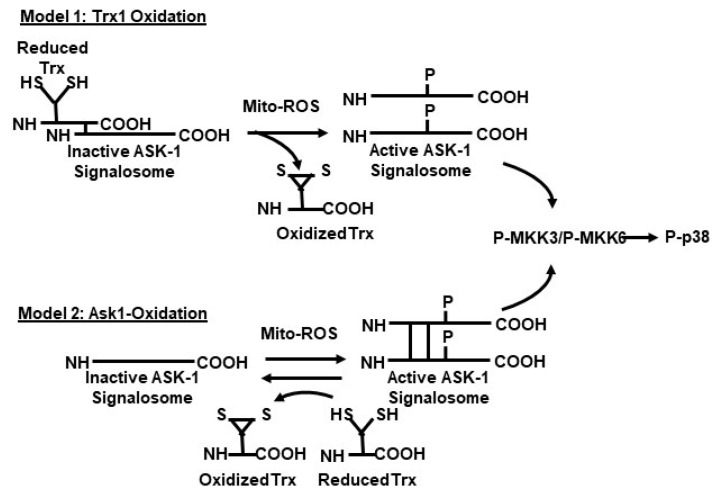
Two proposed model of the mechanism of activation of the ASK1 ROS-Sensory complex **Model 1: The Tri-oxidation model**—The (SH)_2_Trx-ASK1 complex serves as a negative regulator of ASK1 and the association of (SH)_2_ Trx with ASK1 maintains the signalosome in an inactive state. The signalosome is activated by dissociation of oxidized Tri. **Model 2: ASK-oxidation model**—ASK1 Monomers are activated by ROS-mediated multimerization (disulfide bond formation). Inactivation of ASK1 is mediated by (SH)_2_Trx-mediated reduction of the disulfide bond and ASK monomerization.

**Figure 5 cells-08-01383-f005:**
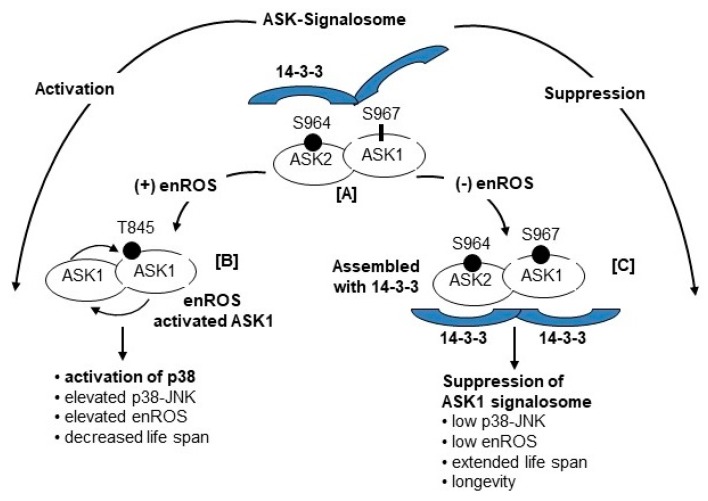
Regulatory process at the C-terminus14-3-3 domain of ASK1-ASK2-responses to ROS. [**A**] The heterodimer of P-ASK2-Ser^964^-ASK1 is bound to 14-3-3 via P-ASK2-Ser^964^; [**B**] Under conditions of elevated ROS, p38 is activated via the formation of the ASK1-P-ASK1-Thr^845^ complex which is released from 14-3-3; [**C**] Under conditions of decreased ROS[(−)ROS] the P-ASK2-Ser^964^ stimulates the phosphorylation of ASK1-Ser^967^ to form the P-ASK2-Ser^964^-P-ASK1-Ser^967^-14-3-3 complex which suppresses the ASK2-ASK1 signalosome and activation. Activation also involves the recruitment of C-terminal regulatory proteins AIP and HIPK1.

**Figure 6 cells-08-01383-f006:**
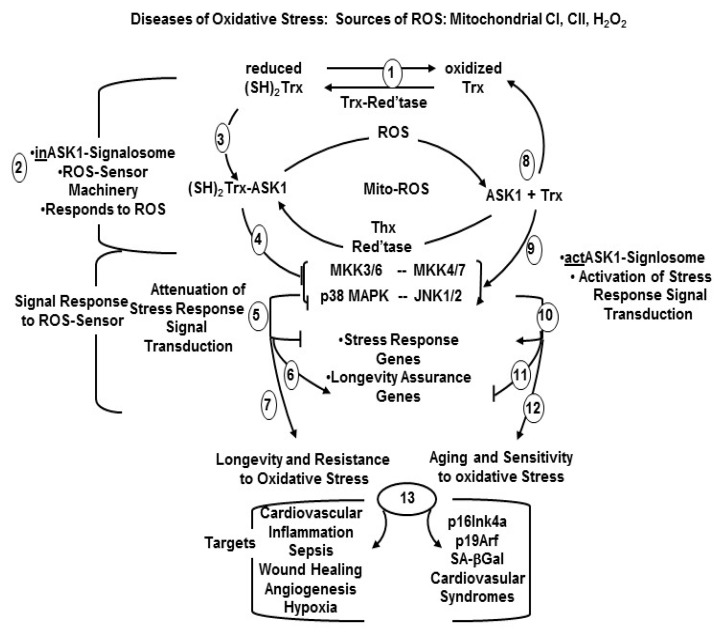
The pathways that promote longevity and resistance to oxidative stress vs. aging and sensitivity to oxidative stress. (1) Thioredoxin reductase plays a major role in balancing the overall cellular oxidation-reduction homeostasis; (3) Reduced Thx binds to the amino terminal of ASK1 to form the inhibited ASK1-signalosome; (4) the inASK1-signalosome attenuates the stress response pathway; (5,6) the stress response genes are attenuated; (7) the longevity response genes re activated; (8) Oxidized Thx is reduced by Thx reductase; (9) The actASK1-signalosome activates the stress response signaling pathway which; (10, 11, 12) activates stress response genes and attenuates longevity assurance genes; (13) the targets of the attenuated stress response pathway favor longevity and resistance to oxidative stress; (13) the act Signalosome promotes aging and sensitivity to oxidative stress and cardiovascular syndromes.

**Figure 7 cells-08-01383-f007:**
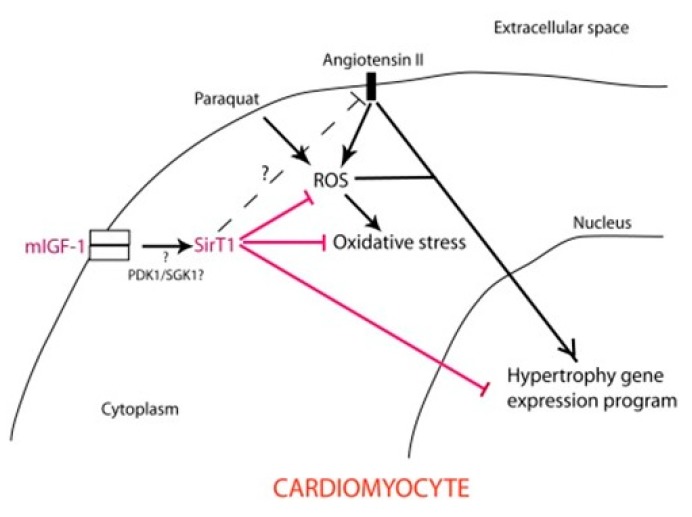
Simplified scheme illustrating the role of mIGF-1-induced SirT activity in protection against Ang ‖- and PQ-mediated oxidative stress and hypertrophy in cardiomyocytes. The IGF-1-mediated induction of SirT-1 activity plays a protective role against Ang‖ and PQ-mediated oxidative stress and hypertrophy in cardiomyocytes. SIRT-1 attenuates the ROS generated by exogenous (PQ) and endogenous (Ang‖) factors thus down regulating the stress response signaling pathways activated by the ASK1-signalosome. From: Manlio Vinciguerra, et al., (2010) Local IGF1 isoform protects cardiomyocytes from hypertrophic and oxidative stress via SirT1 Activity. *Aging* 2, 1075–1094.

**Figure 8 cells-08-01383-f008:**
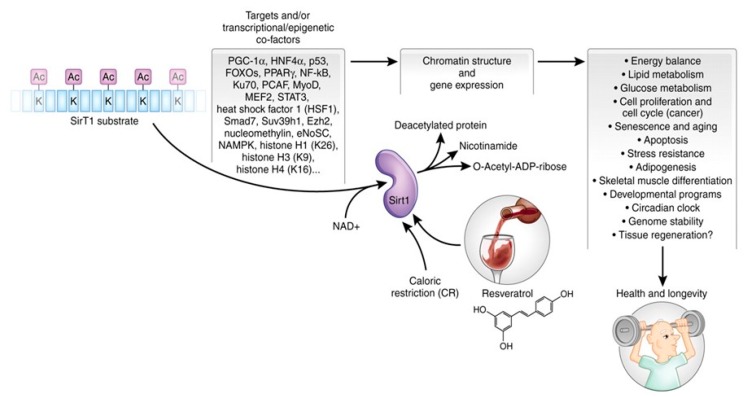
The enzymatic reactions carried out by SirT1, its targets, including transcriptional co-factors, and dependent biological process. SirT1 protein substrate(s) is represented as a string of blue rectangles, with acetylated (Ac) lysine (k) residues. Using NAD^+^ as a co-factor, SirT1 can deacetylate histones, and nuclear and cytoplasmic proteins on specific K residues. This reaction generates a deacetylated protein, nicotinamide and *O*-acetyl-ADP-ribose (OAADPR). SirT1 activity can be enhanced by caloric restriction (CR) and by the polyphenol resveratrol, affecting multiple developmental, physiological and pathological processes, and ultimately favoring health and increasing longevity. From: Vinciguerra M. et al. *Dis. Model. Mech*. 2010, 3, 298–303.

**Figure 9 cells-08-01383-f009:**
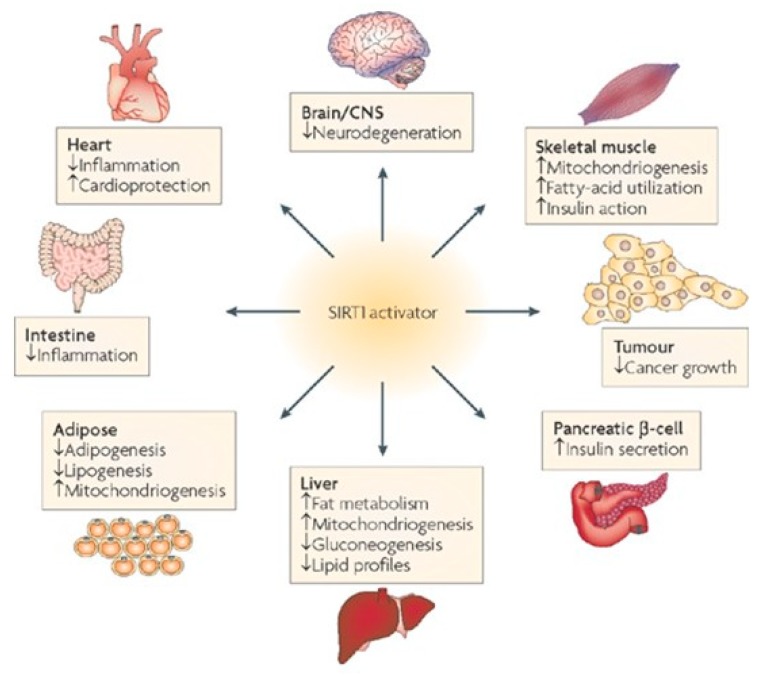
Multiple target organs in which SIRT1 activators can potentially affect the treatment of diseases of ageing. From: Sirtuins—novel therapeutic target to treat age-associated diseases Siva Lavu, et al. (2008) *Nature Review Drug Discovery* 7, 841–853, doi:10.1038/nrd2665.

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
