# Peer review of "The Role of Signaling Pathways of Inflammation and Oxidative Stress in Development of Senescence and Aging Phenotypes in Cardiovascular Disease"

_cells, 2019, doi:10.3390/cells8111383_

Round 1

Reviewer 1 Report

This manuscript reviews the role of the ASK signalosome in senescence and oxidative stress-related diseases. Although the review focuses mainly on senescence, there is no explanation of what senescence is at the molecular level. There is also a lot of repetition in the different section, which could be improved by combining different sections in which the same data is discussed. Detailed comments are as follow: 1) Some sentences are very long. For example line 91-94 and lines 97-100 2) Line 105: remove the “n” from “an ROS” 3) Section 2, lines 107-121. This is very general paragraph. What are the protein-protein interactions, conformational changes , etc…that disseminate ROS? There is a lot of wording in these sentences, but nothing concrete. 4) Line 148. What is the meaning of the regulation of the level of expression? Does the ASK complex regulates transcription of the downstream effectors? Or it is just increasing activity. 5) Are p38 MAPK/JNK kinases part of the ASK complex? 6) Figure 4 could be improved. It is hard to see how the complex is regulated. 7) Line 160: Figure should be Fig 3? There are no regulatory domains depicted in Fig 4 8) Line 166: which are the signals distributed by the ASK complex? 9) Lines 167-168: How does p38MAPK promote senescence, aging and diseases of oxidative stress? 10) Is the ASK complex in the cytosol and in the mitochondria? 11) Line 169: Correct signalosome 12) Line 188, Correct “or example” 13) Section 7, lines 204-219: This is repetition of the previous sections. No new information is discussed in this section. 14) Titles of sections 7 and 8 are the same, but phased in different ways 15) Line 267: define Wip1. Is this a specific phosphatase for p38MAPK? Is the P in front of p38MAPK indicating phosphorylation? 16) Line 271: correct p16 17) Pages 318-319: correct formatting 18) Figure 8, 9 and 10 were taken from other reviews. This is not acceptable. 19) Line 437: PGC1 refers to PGC1alpha? 20) Finally, better models of the structure of the ASK complex and regulation are needed.

Reviewer 2 Report

The review by Papaconstantinou is a thorough and complete overview of the mechanisms involved in the development of senescence and aging processes that lead to cardiovascular diseases. In particular the author proposes that the ASK1-signalosome regulates the progression of cardiovascular diseases. The critical description of the several aspects of this topic stimulates the reader attention to focus on the signalling pathways that originate from the ASK1-signalosome, whose activation is triggered by ROS production. ROS signals are responsible of senescence, aging and cardiovascular diseases.

I have just a note on the use of “we”, “our”, considering that there is only one author in this review. It is, in my opinion, simply a question of style.

I found these two typo errors:

Line 229 “The endogenous levels of the inhibitory ASK1-sigalosome”

Line 626 “response genes associated with aging. Our proposed mechanism implies that the ASK1-siganalosome”

Round 2

Reviewer 1 Report

Although the author responded to many previous critics, a couple of concerns still need to be addressed.

In response the critic #20: “Better models of the structure of the ASK complex and regulation are needed, the author responded “I agree”, however figures are the same, as previous submission. No changes were made to figures in response to this previous critic.

In response to the critic related to the explanation of senescence, the author responded with a series of reviews, which is not addressing the critic. The reader should have all the information needed to understand the major point of the review. The author focuses in physiologic pathways that promotes senescence without explaining what senescence is and what aspect of senescence is being affected. In line 101, what are the senescent products that affect lifespan? Sentences like this one are not very informative.

Figure 7 shows 13 steps in the model, however, step 2 was not described in the figure legend.
